# Nitrogenase beyond the Resting State: A Structural Perspective

**DOI:** 10.3390/molecules28247952

**Published:** 2023-12-05

**Authors:** Rebeccah A. Warmack, Douglas C. Rees

**Affiliations:** 1Division of Chemistry and Chemical Engineering, California Institute of Technology, Pasadena, CA 91125, USA; 2Howard Hughes Medical Institute, California Institute of Technology, Pasadena, CA 91125, USA

**Keywords:** nitrogen fixation, nitrogenase, iron–sulfur clusters, metalloenzymes, cryo-electron microscopy

## Abstract

Nitrogenases have the remarkable ability to catalyze the reduction of dinitrogen to ammonia under physiological conditions. How does this happen? The current view of the nitrogenase mechanism focuses on the role of hydrides, the binding of dinitrogen in a reductive elimination process coupled to loss of dihydrogen, and the binding of substrates to a binuclear site on the active site cofactor. This review focuses on recent experimental characterizations of turnover relevant forms of the enzyme determined by cryo-electron microscopy and other approaches, and comparison of these forms to the resting state enzyme and the broader family of iron sulfur clusters. Emerging themes include the following: (i) The obligatory coupling of protein and electron transfers does not occur in synthetic and small-molecule iron–sulfur clusters. The coupling of these processes in nitrogenase suggests that they may involve unique features of the cofactor, such as hydride formation on the trigonal prismatic arrangement of irons, protonation of belt sulfurs, and/or protonation of the interstitial carbon. (ii) Both the active site cofactor and protein are dynamic under turnover conditions; the changes are such that more highly reduced forms may differ in key ways from the resting-state structure. Homocitrate appears to play a key role in coupling cofactor and protein dynamics. (iii) Structural asymmetries are observed in nitrogenase under turnover-relevant conditions by cryo-electron microscopy, although the mechanistic relevance of these states (such as half-of-sites reactivity) remains to be established.

## 1. Introduction

An essential element for life, nitrogen can be a limiting bioavailable nutrient. Indeed, the development of the industrial Haber–Bosch process in the early 1900s to “fix” atmospheric dinitrogen via reduction in ammonia helped drive the increase in the global population of ~6 billion people over the last century [1]. To overcome the stability of dinitrogen, the Haber–Bosch process utilizes an iron catalyst operating at temperatures of ~500 °C, with high pressures required to shift the equilibrium towards ammonia formation. Under these conditions, molecular dinitrogen dissociates into nitrogen atoms stabilized on the Fe surface that then combine with hydrogen atoms to generate ammonia [2]. The contrast between the “brute-force” nature of the Haber–Bosch process and the process of biological nitrogen fixation catalyzed by the enzyme nitrogenase is particularly striking. The nitrogenase mechanism has found a way to finesse dinitrogen reduction by balancing the generation of a highly activated state that can react with dinitrogen without self-destruction through inactivating reactions with the surrounding protein matrix, the aqueous solvent, and the broader cellular environment. Understanding this paradox between stability and reactivity is key to establishing the nitrogenase mechanism. Beginning with an introduction to the resting state of nitrogenase, this review focuses on the experimental characterization of forms of nitrogenase beyond the resting state. It discusses emerging cryoEM studies that can now be evaluated in the light of decades of biochemical, molecular biology, crystallographic, spectroscopic, and computational studies of the nitrogenase MoFe- and Fe-proteins to provide a fresh perspective on catalytically relevant states of nitrogenase and how they may differ from the resting state.

## 2. Overview of Nitrogenase Structure and Mechanism

### 2.1. Nitrogenase Resting State Structure

As a reference point, we start with the description of the resting states of the nitrogenase proteins and their constituent metalloclusters [3,4,5,6,7,8]. The best-studied molybdenum nitrogenase consists of two component proteins: the molybdenum iron (MoFe-) protein and the iron (Fe-) protein. The MoFe-protein contains the active site for dinitrogen reduction; it exists as an α_2_β_2_ tetramer containing two copies of two unusual metalloclusters: the P-cluster, and the FeMo-cofactor (Figure 1). The α- and β-subunits are homologous, but they occupy distinct positions in the tetramer which can be considered formed from a pair of αβ-dimers. While the emphasis of this review will be on the MoFe-protein, we note two homologous alternative nitrogenases have been characterized: the vanadium iron (VFe-) and iron only (FeFe-) proteins [9,10]. These proteins are both hexamers, with two G-subunits in addition to the conserved α_2_β_2_ core. The Fe-protein is a γ_2_-homodimer with a [4Fe:4S] cluster that supports the ATP-dependent transfer of electrons to the MoFe-protein from ferredoxins and flavodoxins (in vivo) or chemical reductants such as dithionite in vitro. The Fe-protein adopts a common nucleotide binding protein fold and is a member of a class of dimeric nucleoside triphosphatases that are widely distributed throughout living systems [11].

While the FeMo-cofactor and P-cluster each contain eight metals, they have distinctive structures that must reflect their respective functions, albeit in ways that are not yet fully understood. The FeMo-cofactor has the composition [7Fe:1Mo:9S:C]-R-homocitrate organized around a trigonal prismatic arrangement of six Fe ions surrounding an interstitial carbon (Figure 1b). The three pairs of Fe in the trigonal prism are bridged by the so-called “belt sulfurs” that are two-coordinate. A remarkable feature of the FeMo-cofactor is that it is coordinated to the MoFe-protein through only two protein ligands: α-Cys275 to the lone Fe not in the trigonal prism and α-His442 coordinated to the Mo. The Mo environment is completed by bidentate coordination of the R-homocitrate (HCA) ligand. None of the six Fe in the trigonal prism are coordinated by protein ligands in the resting state. The FeMo-cofactor is bound at the interface between the three domains of the α-subunit. While the inorganic component of the cofactor is effectively dehydrated in the resting structures, HCA is surrounded by a pool of buried water molecules.

The P-cluster, with composition [8Fe:7S], is positioned at the interface between the homologous α- and β-subunits (Figure 1b). Three distinctive structures have been identified, assigned to three different oxidation states [12,13]. In the all-ferrous P^N^ state, a hexacoordinate S bridges two [4Fe:3S] subclusters; the 8 Fe atoms are each coordinated by a cysteine sidechain, such that four cysteines (α-Cys62, β-Cys70, α-Cys154, β-Cys153) coordinate single irons; while two cysteines (α-Cys88, β-Cys95) bridge two irons. Structural evidence has shown that in the P^1+^ state, the coordination environment of Fe6 switches from a β-Cys153 to the β-Ser188 sidechain, while additionally, in the P^2+^ state, Fe5 changes from coordinating the thiol of α-Cys88 to the amide backbone of the same residue [13,14]. The two-fold axis that approximately relates to the two subclusters superimposes closely with the two-fold axis approximately relating a pair of α- and β-subunits. The FeMo-cofactor and P-cluster within a given αβ-dimer are separated by about 14 Å, while the metalloclusters in different αβ-dimers are separated by ~65 Å. Consequently, the two active sites have generally been considered to function independently, although evidence for both positive and negative interdimer cooperativity has been presented [15,16,17,18] (discussed in more detail below).

The [4Fe:4S] cluster of the Fe-protein spans its dimer interface, coordinated to γ-Cys97 and γ-Cys132 of each subunit. This cluster has the unusual property relative to other [4Fe:4S] clusters in that it can stably adopt 2+, 1+, and 0 oxidation states [19,20], although the 2+ and 1+ states are thought to be primarily relevant to nitrogen reduction. The Fe-protein is a P-loop NTPase, and binding of nucleotide by its namesake motif is coupled to changes in the cluster through the Switch II motif [11]. The nucleotide-bound state of the Fe-protein also affects the detailed structural interactions with the MoFe-protein [21,22,23,24]. While structures for nucleotide free and ADP-bound Fe-protein are available [25,26,27], an ATP-bound structure of the free Fe-protein has not yet been determined.

### 2.2. The Lowe–Thorneley Model of Nitrogen Reduction

Based on studies of the Klebsiella pneumoniae nitrogenase proteins, Lowe and Thorneley developed a kinetic model for dinitrogen reduction (Figure 2) involving a sequence of increasingly reduced states of the MoFe-protein (designated E_n_, where n is the number of electrons transferred to the active site from the Fe-protein, starting from the resting E_0_ state) coupled to the ATP-dependent transfer of electrons from the Fe-protein (reviewed in [28]). Since its formulation, the Lowe–Thorneley model has provided the dominant conceptual framework for viewing the kinetic nitrogenase mechanism. Important features of this model are as follows: The MoFe-protein must be reduced by the Fe-protein by three to four reducing equivalents prior to dinitrogen binding [29]. Other substrates and inhibitors (C_2_H_2_, CO, CN^−^) can bind to less highly reduced forms, including potentially the resting E_0_ state [29,30,31,32].The rate-determining step is not involved with substrate reduction chemistry; rather, was identified as dissociation of the MoFe-protein–Fe-protein complex [33]. Subsequent studies assigned the rate determining step to Pi dissociation [34]. An important consequence of this property is that the flux of electrons through the MoFe-protein is independent of the substrate being reduced [3]. Furthermore, in the absence of other reducible substrates, protons are reduced to H_2_ at the same rate of electron flow.The kinetic studies did not independently identify the proposed E_n_N_x_H_y_ intermediates often depicted in the Lowe–Thorneley scheme (Figure 2). To date, spectroscopic and biochemical studies by Hoffman, Seefeldt, Dean, and colleagues have characterized several of the intermediates proposed in the reduction of N_2_, including a species with the stoichiometry of “2N2H” for the E_4_ state. The structure “2N2H” fragment is unknown but is consistent with either a diazene-level intermediate or dinitrogen dihydride species [35]. This collaboration has also assigned the E_7_ and E_8_ states as NH_2_- and NH_3_-bound FeMo-cofactors states, respectively (see [36]). It is important to note, however, that no definitive evidence for any on-path, partially reduced NN intermediate has been presented [37]. The lack of characterization includes the E_5_ and E_6_ intermediates predicted to have reduced N–N bonds, presumably because of challenges in the spectroscopic characterization, but it should not be completely dismissed that species with reduced N–N bonds do not exist.In addition to the electron transfer role, Fe-protein bound to the MoFe-protein prevents substrate binding and product release, as only free MoFe-protein can bind substrate and release products [28]. As emphasized by Thorneley and Lowe [38], this is an essential role for the Fe-protein as it enables the MoFe-protein to function as a nitrogenase and not just a hydrogenase. Hence, the term “dinitrogenase reductase”, sometimes used for the Fe-protein, is inappropriate as the Fe-protein plays an important role in the nitrogenase mechanism beyond solely serving as an electron donor.

These studies emphasize that the resting state, which has been the primary state structurally characterized, is an inactive state and must be activated prior to N_2_ reduction.

### 2.3. Binding of Ligands to the FeMo-Cofactor

The senior author of this review recalls an expectation that elucidation of the FeMo-cofactor structure would, if not reveal the mechanism of dinitrogen reduction, at least inform the site of dinitrogen binding. This expectation was not realized for two basic reasons:The early model proposed by the Rees group lacked the interstitial ligand [39].The resting state of the FeMo-cofactor does not bind N_2_ [33].

Over time, a picture of how substrates can bind to the FeMo-cofactor has emerged based on multiple lines of experimental evidence:A binding site near Fe6 was proposed by Dean and co-workers based on the consequences of mutations involving residues α-His195 and α-Val70 [32,40,41,42].Spectroscopic studies revealed the presence of hydrides, bound N_2_, and mononuclear N under turnover conditions (reviewed in ref. [36]). These studies provide unique experimental insights into key species generated under turnover conditions.A stably trapped CO-containing species generated under turnover conditions has been crystallographically characterized [43]. CO was observed to bridge Fe2 and Fe6, which form one edge of the trigonal prism, displacing belt sulfur S2B. Subsequent studies pressurizing these crystals under CO revealed the binding of a second CO, in an end-on fashion, to Fe6 [44].Under turnover in the presence of selenocyanate, Se was also found to bind predominantly at the S2B position in a catalysis-dependent fashion [45]. Using the incorporated Se as a site specific probe, the exchangeability of all three belt sulfur sites under turnover conditions was established. This was demonstrated by showing the migration of selenium between these sites under turnover, with eventual efflux of the Se from the cofactor. Interestingly, upon reaction with CO under turnover conditions, rather than displacing Se at S2B, Se predominantly moves to other belt sulfur positions [45,46]. These observations have provided direct evidence that the FeMo-cofactor can rearrange during catalysis.The VFe-protein solved by Einsle and coworkers [47] provides a fascinating comparison to the MoFe-protein as the S3A belt sulfur bridging Fe4 and Fe5 is replaced with a tetra-atomic ligand assigned as carbonate. Intriguingly, a second form of the VFe-protein with distinct EPR features was identified in which the S2B belt sulfur was occupied with a protonated light atom, likely NH or OH [48]. This replacement was accompanied by a rearrangement of the conserved α-Gln191 (α-Gln176 in the VFe-protein) to generate a binding site for the presumably displaced S2B species. As with the MoFe-protein, the S2B site in the VFe-protein can be displaced with CO under turnover conditions, and a second CO may also be bound to Fe6 [49]. These observations have lead to the formulation of a detailed mechanistic proposal for substrate reduction by nitrogenase [49,50].The structure of the iron-only Fe-nitrogenase solved by the Einsle group [51] also supports lability of the S2B belt sulfur in the FeFe-cofactor, as evidenced by two alternative conformations for α-Gln176 (corresponding to α-Gln191 of the MoFe-protein) and a binding site for the displaced sulfide.Intriguing biochemical and structural data has been presented for nitrogen containing species bound to the FeMo-cofactor under conditions of sulfur and electron depletion [52]. Reflecting the difficulty of identifying small, light atom ligands bound to an electron density metallocluster, this interpretation has been challenged [53,54]. While this challenge has been rebutted [55,56], our examination of the supporting crystallographic data does not definitively support N_2_ displacement of belt sulfurs, leaving unclear the structural basis of the biochemical observations.

These observations of ligand binding to the nitrogenase cofactor have two important mechanistic consequences for events occurring beyond the E_0_ resting state:The identification of hydrides on the cofactor provides a mechanism whereby transfer of electrons to the cofactor can proceed without an accumulation of electrons; this would presumably allow for electron transfer from the Fe-protein to the active site to occur at a nearly constant potential [35,56,57]. The reductive elimination of H_2_ from the four-electron-reduced E_4_ state was recognized [58,59,60] as a mechanism to generate a two-electron-reduced form of the cofactor that is transiently activated for the binding and reduction of N_2_. This step represents a mechanism whereby a reactive species would be transiently generated under turnover; in the absence of N_2_, H_2_ would be evolved, returning the active site to a less reactive state.A central role is highlighted for Fe2 and Fe6 forming a binuclear substrate binding site in the trigonal prism of the active site cofactor [43,44,45,48,49,51].The belt sulfur S2B can be displaced by exogenous ligands under turnover conditions [43], and evidence for the interchange of belt sulfur positions has been presented [45].

### 2.4. Proton Coupled Electron Transfer Reactions Relevant to Nitrogenase

The reduction of substrates by nitrogenase is orchestrated through the delivery of electrons and protons to the active site. Starting from the resting state, the progression through the more highly reduced states of the catalytic cycle is coupled to ATP-dependent electron transfer to the FeMo-cofactor from the Fe-protein [61,62]. Protons arrive through various pathways that have been proposed to link the active site to the protein surface (see [63,64,65]). While the overall ratio of 1 proton per 1 electron is set by the reaction stoichiometry, many mechanistic descriptions of nitrogenase indicate that substrate reduction proceeds through a sequence of steps such that each electron transfer is stoichiometrically coupled to a proton transfer [36,62]. As explored below, however, experimental evidence for this detailed level of coupling is sparse. We start by examining the coupling of proton and electron transfer to synthetic iron sulfur (FeS) clusters and FeS clusters in small proteins.

#### 2.4.1. Proton Coupled Electron Transfer Reactions of FeS Clusters

The evidence is quite clear that redox reactions involving synthetic FeS centers are generally not accompanied by obligatory proton transfers to the FeS center. This was established from the earliest characterization of synthetic 1-Fe (Fe:4(SR)^z^), 2-Fe (2Fe:2S:4(SR)^z^), and 4-Fe (4Fe:4S:4(SR)^z^) compounds by X-ray crystallography in multiple oxidation states (see ref. [66]), demonstrating that cluster reduction does not require cluster protonation. A beautiful example of this is the recent characterization [67] of a series of [4Fe:4S] clusters spanning the five oxidation states ranging from all ferrous to all ferric irons. While the cluster was not observed to be protonated in any of these oxidation states, intriguingly the more highly reduced states were stabilized by interactions with K^+^ ions. 

Due to the typically lower resolution of protein crystal structures relative to small molecule crystal structures, the evidence against cluster protonation in protein bound FeS clusters is less extensive, but there is still decisive evidence against cluster protonation in at least two cases of protein structures: perdeuterated *Pyrococcus furiosus* rubredoxin solved by neutron diffraction in either oxidized or reduced forms [68] and the *Thermochromatium tepidum* HiPIP solved to 0.8 Å resolution at pH 4.5 by X-ray crystallography, with a neutron diffraction structure of the oxidized state at 1.2 Å resolution [69,70]. 

Not only can FeS cluster reduction occur without protonation, but cluster protonation can occur without reduction. This process has been best characterized from the acid catalyzed kinetics of the ligand substitution reactions of the terminal alkyl thiolates in synthetic [4Fe:4S] clusters [71,72,73]. The pK for this process has been found to be in a physiologically relevant range and to be largely independent of the detailed nature of the reaction components, indicating that protonation involves a common group, suggested (but not proven) to be the µ_3_ bridging sulfurs in the cluster [74]. Of tremendous practical significance in working with FeS proteins, cluster protonation has also been implicated as an early step in the decomposition of FeS clusters at low pH [75].

In certain cases, however, electron and proton transfers are found to be coupled. Evidence for such coupling has been provided by the pH dependence of the reduction potentials for the generation of more highly reduced states of FeS clusters in small proteins. For the stoichiometric coupling of proton and electron transfers, the reduction potential will shift by −59 m/n mV per pH unit, where m and n are the number of protons and electrons, respectively, associated with a reduction reaction:A_ox_ + mH^+^ + ne^−^ → A_red._

Examples of such behavior have been reported for the reductions of the [2Fe:2S] Rieske cluster to the all-ferrous +0 state [76], the *Azotobacter vinelandii* ferredoxin [3Fe:4S] cluster to the +0 state [77], and the *Sulfolobus acidocaldarius* ferredoxin [3Fe:4S] cluster to the all ferrous −2 state [78]. While these studies cannot identify the proton binding site (including whether the cluster itself is protonated), the favored hypothesis in each case proposed that protons bind to μ_2_ sulfides. 

In general, though, the reduction potentials of FeS proteins are either pH independent, or the pH dependence is much less pronounced than shifting by −59 m/n mV/pH (an early tabulation is provided in [79]). The latter cases are typically attributed to either redox state dependent shifts in the pKas of protonatable residues or to pH-dependent protein conformational changes (see [79]). The pKa is greater in the reduced than oxidized state, so increasing pH stabilizes the oxidized form and the reduction potential decreases with pH. As these residues may not be immediately adjacent to the redox center, the shifts in reduction potential tend to be more modest than for direct cluster protonation.

An interesting situation arises when the titratable group is a ligand to the FeS cluster. Cysteine ligands are sufficiently acidic that they typically coordinate as the deprotonated thiolate anion. An exception to this statement is provided by a [NiFe] hydrogenase in the Ni-R state with a protonated cysteine ligand to the Ni identified by X-ray crystallography at 0.89 Å resolution [80]. Upon replacement of cysteine cluster ligands with more basic residues, pronounced pH dependences of the reduction potential can appear. Examples of this phenomenon are provided by the *Clostridium pasteurianum* rubredoxin and several bacterial [2Fe:2S] Rieske clusters, where substitution of cysteine ligands by serine [81] and histidine [82], respectively, confer a strong pH dependence not present in their counterparts with all four cysteine ligands.

#### 2.4.2. Proton Coupled Electron Transfer to the Nitrogenase Metalloclusters

Throughout the catalytic cycle of nitrogen reduction by nitrogenase, metalloclusters within the Fe-protein and the MoFe-protein adopt various oxidation states. Protonation of the nitrogenase metalloclusters in these states has been challenging to experimentally address due to the complexity of these centers and the reaction mechanism. In addition to the cluster sulfides, potentially protonatable groups include the amino acid ligands to the nitrogenase metalloclusters (histidine to the FeMo-cofactor and an amide N and/or serine in certain oxidation states of the P-cluster), the interstitial carbon, and homocitrate in the various FeX-cofactors; additionally, carbonate provides an unusual opportunity for ligand protonation in the FeV-cofactor [47]. In addition to these proton binding sites, the protonation of cluster irons would result in hydride formation. As discussed below, the available evidence indicates that the clusters are not directly protonated in the resting state, although proton transfer to the active site certainly occurs during the catalytic cycle.

P-cluster: As noted above, the MoFe-protein P-cluster exhibits three distinct reversibly oxidized structural states designated the P^N^, P^1+^, and P^2+^ states [13]. Since the protein ligands to the P-cluster are potentially protonatable, it might be anticipated that the P-cluster reduction potentials are pH dependent, as is observed [83]. The P^2+^/P^1+^ potential exhibits a −53 mV/pH shift, presumably due to the protonation of the amide nitrogen upon dissociation from Fe5. In contrast, the P^1+^/P^N^ potential is pH invariant, suggestive that the β-Ser188 side chain may be protonated, irrespective of whether or not it is coordinated to Fe6.

An intriguing aspect of the P-cluster is that some substitutions, or even deletions, of the cluster cysteine ligands can be generated that are still able to grow diazotrophically, including substitutions at α-Cys88 that bridges the two subclusters [84,85]. Structures from the Tezcan group [86,87,88] of point mutations in the oxygenic residues surrounding the P-cluster demonstrated that the cluster could become compositionally labile upon oxidation and lose Fe centers while still retaining near wild-type levels reduction of nitrogen. Based on these observations, the authors proposed that inherent lability of the P-cluster may play a role in catalysis. These results are suggestive that a more conformationally complicated pathway for electron transfer may be involved beyond interconversion between the most extensively characterized P^N^, P^1+^, and P^2+^ states.

FeMo-cofactor: Potential sites for protonation of the FeMo-cofactor include the µ_2_ belt sulfurs, the µ_3_ cluster sulfurs, homocitrate, the interstitial carbon, and the protein ligands; protonation of irons will yield hydrides. Solvent molecules in the vicinity of the cofactor may also participate in proton transfer reactions. At the current resolution of the nitrogenase structures determined by X-ray crystallography (~1 Å), unambiguous assignment of hydrogens is not possible. Re-refinement of the 3U7Q crystal structure testing various potential protonation states has established that the unprotonated structure best fits the experimental data [89]. Moreover, there are no examples of exclusive hydrogen bond acceptors (such as carbonyl oxygens) near the cofactor in the resting state ~pH 8 that could provide evidence for hydrogen bond donors (i.e.,—protons on the cofactor). The characteristic EPR spectra arising from the FeMo-cofactor in the resting state is pH dependent [90,91], demonstrating that protonation events can influence the properties of the cofactor, but not specifically that protonation of the cluster is occurring. Structural analyses of *A. vinelandii* MoFe-protein and *C. pasteurianum* MoFe-protein at low pH revealed conformational changes near the FeMo-cofactor, suggesting that belt sulfurs S3A and S5A are potential protonation sites [91]. Computational analysis has indicated, however, that protonation of these two belt sulfurs should be energetically less favorable than for S2B [89]. While the observed structural and electronic low-pH changes were correlated and reversible, somewhat surprisingly, the structural rearrangements differ between the two MoFe-proteins.

Details of the protonation sites in more highly reduced states of the cofactor have been challenging to define due to the difficulties in obtaining high resolution structural data on these forms. Protonation sites on the cofactor are generally considered as involving cluster sulfides and/or Fe (leading to hydride formation), although reduction of the interstitial carbon has also been proposed [92,93]. Spectroscopic methods provide powerful approaches to investigate hydrogens bound to the cofactor. Pioneering ENDOR studies by Hoffman and coworkers have identified Fe-hydride species in both the E_2_ and E_4_ states [94,95,96]; corresponding studies in the E_1_ and E_3_ states are not possible, however, due to their non-Kramers spin states. The formation of Fe-hydride species may reflect the role of Fe-centered reductions of the cofactor during turnover, while the Mo has been reported to remain redox innocent [97,98]. 

#### 2.4.3. Computational Studies of the FeMo-Cofactor beyond the Resting State

A detailed understanding of the consequences of proton and electron transfers to the active site of nitrogenase is critical for defining the mechanism of substrate reduction in molecular detail. A key advantage of computational quantum mechanical studies is that they can directly simulate a single species and hence evaluate the potential mechanistic relevance of proposed intermediates. These studies are therefore indispensable for probing possible reaction pathways for substrate reduction. The challenge, however, is that the results can depend significantly on the computational details, particularly the choice of starting model and the functional used in the density functional calculations. The choice of model involves which atoms to include in the calculation from the metalloclusters (with the associated issues of oxidation and spin states), the surrounding protein and solvent, the protonation states of potentially ionizable species, and the locations of species that might arise under turnover conditions, such as hydrides and H_2_. As a consequence, hundreds of models may need to be evaluated (see [89,99]), particularly for more highly reduced states. Even with the same model, the results can depend significantly on the computational details of the density functional methods [100]. To the non-expert, while there is no easy way to assess the accuracy of a given model, there are distinctive features for some of the predicted structures that may serve as guides for interpreting future structural data.

It is important to recognize that these calculations are largely based on the coupled transport of H^+^ and e^−^ to the cofactor driving the transition between E_n_ states. As a consequence, the charge of the system remains constant during turnover, which has important consequences for the calculation to minimize the consequences of long-range electrostatic effects on the energetics (see [101]). As discussed above (Section 2.4.1), the evidence that proton transfer is an obligatory feature of FeS cluster reduction is weak, however. This assumption should be consequently reevaluated since the irons in the cofactor are on average partially oxidized and therefore quite distinct from the all-ferrous forms of FeS clusters where coupled proton transfer has been reported (Section 2.4.1). 

The candidate protonation sites most commonly explored in the FeMo-cofactor are the cluster sulfurs (both the belt µ_2_ species and the µ_3_ species in the partial cubanes), Fe sites (and the resulting hydrides that can be further protonated to form H_2_), the interstitial carbon, homocitrate, and surrounding protein residues. The combinatorics are such that a large number of species can potentially arise, particularly in more highly reduced forms, Reflecting this situation, Dance notes in his comprehensive review of computational studies of the nitrogenase mechanism that “The geometrical and electron structures are too numerous to present here, and it is not possible to conclude a geometrical structure and electronic state for the E_1_H_1_, E_2_H_2_, E_3_H_3_ and E_4_H_4_ intermediate” [102]. Consequently, the following discussion is intended to be representative and can only cover a small set of outcomes reported for these states.

Protonation of cluster sulfurs: A common feature of computational studies is that the early reduction steps are associated with protonation of a belt (µ_2_) sulfur, typically S2B [89,93,99,101,103,104,105,106]. The protonation of other sulfurs has been proposed—both the belt sulfur and µ_3_ sulfurs in the partial cubanes, although the latter are generally found to be less-easily protonated (see discussions in [89] and [105]), as noted in the discussion above on protonation of all-ferrous FeS clusters (Section 2.4.1). Protonation of a cluster sulfur can lead to dissociation of that species from one (or more) Fe; indeed, several studies have proposed that following the transfer of two protons to S2B, this species dissociates from Fe2 and Fe6 [101,103].

Hydride formation: Protonation of the iron sites can lead to hydride formation with accompanying iron oxidation. These species can exist as either terminal or bridging hydrides, most notably the two bridging hydrides experimentally identified by Hoffman and colleagues in the E_4_ intermediate [58,59,60]. Hydride formation has been reported, starting with the E_2_ state [89,99,104,105,107]. The existence of multiple isomers with similar energies, but differing in the detailed locations of the hydrides, suggest that a dynamic equilibrium between alternate hydride positions may occur on the FeMo-cofactor during turnover [105].

Protonation of the interstitial carbon: Several studies report the observation that the transfer of one or more protons to the interstitial carbon during the catalytic cycle can be energetically feasible relative to hydride formation [92,93,108]. Other studies note that these results depend strongly on the choice of functional (see [100]). Hydrogenation of the interstitial carbon can substantially perturb the FeMo-cofactor geometry, potentially opening up the cofactor interior for substrate interactions. 

One implication of the computational studies is that proton transfer to the active site of nitrogenase likely involves the unique features of the cofactor, such as hydride formation on the trigonal prismatic arrangement of irons, protonation of the belt sulfurs spanning these irons, and/or protonation of the interstitial carbon. As these features are not well-represented in the best characterized synthetic FeS clusters, this could provide a rationalization for why the coupling of protein and electron transfer in nitrogenase may occur even though it does not appear to be an obligatory part of the redox chemistry of the synthetic clusters. Computational studies also provide some hope that it may be possible to establish the presence of some of these intermediate states using structural methods such as electron microscopy or X-ray crystallography. While it is unlikely that hydrogens or hydrides will be observed by these methods (unless very high resolution data can be obtained (as for the [NiFe] hydrogenase mentioned above [80]), the types of changes proposed for certain intermediates in Fe–S bonding and/or interstitial carbon protonation should be observable, provided sufficient resolution of the cofactor region can be achieved.

## 3. Structural Evidence for Dynamic Nitrogenase States

While structural biology provides an avenue for visualizing conformational changes that might activate the FeMo-cofactor for substrate reduction, intermediate states of the nitrogenase mechanism have proved difficult to capture for crystallographic studies in part due to their transient nature, low abundance, and the challenges of characterizing small ligands bound to electron dense metalloclusters. Recent cryoEM structures of turnover-relevant states have pointed to a role for larger conformational flexibility within the MoFe-protein during turnover. Moreover, structural studies of mutated and inhibited states of the Fe- and MoFe-proteins have provided important insights into the pliancy of the cofactor and surrounding residues. In this section, we will review the structural evidence from these studies for conformational variability in the nitrogenase polypeptides and the possible relevance for catalysis.

### 3.1. High pH-Inhibited and Inactivated States

The reduction of substrates by nitrogenase is accompanied by the concomitant reduction of protons to dihydrogen. Accumulated reducing equivalents can be lost from the MoFe-protein in the form of dihydrogen, which will slow the accretion of more reduced forms of the enzyme. One way to reduce the rate of formation of dihydrogen is to limit the proton concentration by working at higher pHs [109]. Burgess and Pham studied the effects of pH on substrate reduction and reported that for H_2_ evolution and acetylene (C_2_H_2_) reduction, there is evidence of a group with a pK of about 6.3 that must be deprotonated and a group with a pK of about 9.0 which that be protonated for substrate reduction to occur [110]. Subsequently, Yang et al. reported that reduction of substrates at higher pH results in the irreversible inactivation of the MoFe-protein [109]; this species was found to have a full complement of metals and an increased hydrodynamic radius, while maintaining the ability to interact with the Fe-protein. This turnover-dependent inactivation coupled to a conformational change in the MoFe-protein suggested a mechanism-based transformation occurred under these conditions that could provide insights into nitrogenase catalysis.

As an initial effort to understand the effect of pH on the MoFe-protein structure, crystallographic studies from the Rees lab investigated the effect of pH on the resting state of both the *Clostridium pasteurianum* and *Azotobacter vinelandii* MoFe-proteins [91]. These studies produced crystal structures of *C. pasteurianum* MoFe-protein at pH 5, 6.5, and 8, as well as a structure of *A. vinelandii* MoFe-protein at pH 5. Morrison et al. [91] described alterations in residue α-His274 in the *A. vinelandii* MoFe-protein and residue α-Arg347 in the *C. pasteurianum* MoFe-protein at low pH, suggesting that the conformations of these residues are sensitive to protonation events. Within the low-pH *A. vinelandii* MoFe-protein structure, changes are also seen in residues α-Trp253, α-Phe300, and α-His362 just outside of the active site (Figure 3). A structure of the *A. vinelandii* MoFe-protein at pH 9.5 revealed few structural changes outside a slight displacement of the C1 carboxyl arm of homocitrate [109]. 

Despite the ability to structurally characterize the resting states of *A. vinelandii* and *C. pasteurianum* MoFe-proteins at different pHs, the high-pH turnover-inactivated MoFe-protein (MoFe^Alkaline-inactivated^) state described in Yang et al. remained resistant to crystallization [109]. Recently, Warmack et al., presented a cryoEM structure of the MoFe^Alkaline-inactivated^ state isolated from acetylene reduction reaction mixtures which suggested that homocitrate is lost from the inactivated protein [111]. Coupled with the loss of this moiety, the structure exhibits significant disordering within the α-subunit (Figure 4a), specifically the αIII domain and the active site of the MoFe-protein (Figure 4b,c), but no significant loss of Fe or Mo is observed, indicating that the clusters are still associated with the protein. Interestingly, the MoFe^Alkaline-inactivated^ structure also exhibits altered positioning of α-His274, α-His362, α-Trp253, and α-Phe300, which adopt the same conformations seen in the pH 5 *A. vinelandii* MoFe-protein structure (Figure 4d). Additionally, within the MoFe^Alkaline-inactivated^ state, α-His442 appears to dissociate from the Mo and at least partially occupies a site that joins with α-His274, α-His362, and α-His451 to form a His-quartet and tetrahedral coordination site for an unknown atom, possibly an ion (Figure 4e).

These results establish that the loss of homocitrate can occur under proton-limited turnover conditions. Although the exact function of homocitrate in the nitrogenase turnover mechanism is still enigmatic, suggested roles include involvement in proton transfer to the active site [112] and/or facilitating changes in Mo coordination that could transiently open ligand binding sites [113]. By preparing altered forms of the MoFe-protein with homocitrate substituted with various analogs, Imperial et al. established that the key chemical features required at this site for active nitrogenase are the hydroxyl group, two carboxyl groups, and the *R*-configuration about the chiral center [112]. These stringent requirements for activity bolster the hypothesis that homocitrate plays an important role in nitrogen reduction. The disappearance of this functional group under high-pH turnover conditions may suggest that a deprotonation event occurs which cannot be reversed due to the limited concentration of protons, resulting in an irreversible rearrangement leading to the loss of homocitrate from the protein. Alternatively, the elevated concentration of hydroxide ions in the alkaline solution may result in a nucleophilic attack that displaces the homocitrate.

The active site disordering observed in the cryoEM structures supports the possibility of altered Mo coordination, including dissociation of the α-His442 side chain. This could facilitate repositioning of the FeMo-cofactor while it remains tethered to the protein through the α-Cys275 side chain thiol. While the loss of one HCA ligand has been previously proposed to open up a coordination site on the Mo [114], an alternate possibility might be the binding of a second carboxylate to form a tridentate HCA complex with Mo, coupled to the concomitant displacement of the α-His442 ligand. Indeed, hydroxytricarboxylic acids exhibit a strong preference to form tridentate metal complexes [115]. Displacement of the histidine sidechain would permit repositioning of the FeMo-cofactor, including the possibility of rotation around the Fe1-α-Cys275 sulfhydryl bond. Such behavior could account for the apparent interconversion of the belt sulfur positions under turnover conditions [45]. An alternative would be an intramolecular rearrangement involving internal scrambling of belt sulfurs, but an analysis has noted that such a process would have a high energetic barrier [116]. Whatever the mechanism, the experimentally observed interconversion of belt sulfurs opens the possibility that ligands bound to the cofactor may experience different environments during a catalytic cycle [45,52].

### 3.2. Structural Studies of Nitrogenase from the Deletion Strains ∆nifV and ∆nifB

To assess the relationship between homocitrate and the structural features observed in the MoFe^Alkaline-inactivated^ state, Warmack et al. determined the cryoEM structure of the MoFe-protein isolated from a strain of *A. vinelandii* with a *nifV* deletion, the gene encoding for homocitrate synthase [111]. This structure, termed MoFe^∆nifV^, recapitulates many of the features observed within the MoFe^Alkaline-inactivated^ state, including asymmetric disordering of the αIII domain and shifts in residues α-His274, α-His362, α-Phe300, and α-Trp253 (Figure 5). As seen in the ∆*nifV* structure from *Klebsiella pneumonia* [117]*,* citrate was found to partially occupy the abandoned homocitrate site; however, in contrast to that crystal structure, citrate incorporation was observed asymmetrically within the cryoEM structure. In addition, the *K. pneumonia* ∆*nifV* structure exhibited no changes in the positionings of the α-His274, α-His362, α-Phe300, and α-Trp253 residues. These disparities between the *A. vinelandii* and *K. pneumonia* ∆*nifV* mutant MoFe-proteins may be due to a number of factors, including intrinsic differences between these two distinct proteins, the use of different growth conditions, the differential ability of the organisms to incorporate citrate into the nitrogenase active site, or different structural techniques (X-ray crystallography versus cryoEM). Regardless of the differences between the structures, it is clear from these studies that citrate can be variably incorporated into the active site, and that without the tricarboxylic acid component, the active sites become more disordered. Significantly, despite this disordering, the proteins do not appear to lose the FeMo-cofactor. 

NifB is the radical SAM enzyme catalyzing the critical first step in FeMo-cofactor biosynthesis. Strikingly, the structure [118] of a FeMo-cofactor-deficient MoFe-protein purified from a ∆*nifB* strain (MoFe^∆nifB^) shares many similar features with the MoFe^∆nifV^ and MoFe^Alkaline-inactivated^ structures [111]. This crystal structure displays disordering within the αIII domain and again exhibits rearrangements in the side chains of α-His274, α-His362, α-Phe300, and α-Trp253 residues. Intriguingly, it also exhibits a similar rearrangement in α-His442, as observed in the disordered active site of MoFe^Alkaline-inactivated^. However, the rearrangement of α-His362 differs significantly from all other structures, as within MoFe^∆nifB^, the loop housing this residue rearranges to present α-His362 on the surface of the protein. Based on the structural observations, this residue was proposed to coordinate the FeMo-cofactor to aid in insertion into the apo-MoFe-protein. The significant similarities between the structures of this cofactor-deficient form and the cryoEM structures of MoFe^∆nifV^ and MoFe^Alkaline-inactivated^ might suggest that features such as α-subunit disordering and rearrangements in the His-quartet region and α-Trp253 are correlated with co-factor entry or exit. Interestingly, a subset of the MoFe^∆nifV^ particles analyzed by cryoEM were found to be associated with a novel endogenous binding partner, the nitrogenase associated factor T (NafT) [111], also found on the major cluster of the *nif* operon [7]. Although the function of this binding partner is unknown, it is possible that it acts in a regulatory capacity to recognize cofactor-compromised states of the MoFe-protein.

### 3.3. Structural Studies of Nitrogenase Active Site Point Mutants

Mutagenesis studies of specific residues around the active site of the MoFe-protein have been invaluable for establishing the functional roles of key residues. In this review, we highlight several residues for which structural characterization is available. A mutation of α-Val70 to isoleucine significantly ablated substrate reduction activity, while mutations of α-Val70 to smaller residues allowed for increased activity against larger substrates [119]. The structure of this mutant was solved to 2.3 Å resolution and the authors of this study note that the Cδ1 of α-Ile70 would sterically block substrate access to Fe6 within the FeMo-cofactor (Figure 6) [120]. The reduction in activity upon mutagenesis of this residue to larger amino acids was taken as evidence that the Fe2-Fe3-Fe6-Fe7 face of the co-factor is the substrate binding site.

Several studies have investigated the effects of mutations at sites adjacent to α-Val70, namely, α-His195 and α-Arg96 [121,122]. Mutations at α-His195 were found to eliminate N_2_ reduction activity [42], but the enzyme retained C_2_H_2_ reduction activity. Interestingly, the structure of this mutant protein proved to be nearly identical to the wild-type protein, and changes in the enzymatic activity were attributed to the loss of NH-S hydrogen bonding between α-His195 and S2B of the FeMo-cofactor (Figure 6). In contrast, mutation of α-Arg96 to a glutamine and subsequent saturation of the crystallization mixture with acetylene was reported to result in an acetylene-trapped state of the MoFe-protein, with the structure resolved to 1.7 Å resolution. This structure also exhibits a nearly identical backbone to the wild-type resting-state MoFe-protein with a root mean square displacement (RMSD) of 0.23 Å^2^. The modeled acetylene molecule is positioned 4.2 Å away from Fe6 of the FeMo-cofactor and 2.9 Å from the mutated α-Gln96 side chain (Figure 6). The identification of this density as acetylene was supported by density functional theory (DFT) calculations.

### 3.4. CryoEM Studies of Turnover States

As we have emphasized, capturing intermediate states of nitrogenase, particularly substrate-bound intermediates, is an imposing challenge due to their low abundance and transient nature. As for so many aspects of structural biology, advances in cryoEM are a game changer for these types of studies.

Significant advancements in both hardware and software for cryoEM in the last decade have made it possible to parse low-abundance intermediate states from heterogeneous reaction mixtures [123]. This method has the additional benefit of viewing protein particles trapped in a pseudo-solution state free of possible crystallographic artefacts due to restrictions of rigid lattices. Rutledge et al. employed single-particle cryoEM to analyze steady-state reaction mixtures of the MoFe-protein within high-flux N_2_ reaction mixtures [15]. The authors were able to isolate two states distinguished from these reaction mixtures by nucleotide-binding state. These states were termed ^t/o^Complex-1 and ^t/o^Complex-2, wherein ^t/o^Complex-1 was modeled as a fully ATP-bound state and ^t/o^Complex-2 was a mixed ATP:ADP bounded state. Strikingly, both complexes were 1:1 MoFe-protein:Fe-protein states, and no 2:1 complexes were found in the dataset. The authors proposed that this is suggestive of half-reactivity with the MoFe-protein being able to only bind one Fe-protein at a time during turnover.

Similar features were also observed in the MoFe^Alkaline-inactivated^ state described above [111], including asymmetric disordering of the αIII domain within the MoFe-protein, as well as distorted density around the cofactor and homocitrate within the active site (Figure 7). In addition, changes were observed in the side chains of α-Trp253, α-His274, α-Phe300, and α-His451, as seen in the MoFe^Alkaline-inactivated^ state. As described above, similar changes have been observed in the cofactor-deficient MoFe^∆nifB^, homocitrate-deficient MoFe^∆nifV^, and low pH MoFe-protein structures, indicating that these alterations are not exclusive to turnover. Collectively these structures have emphasized the propensity for conformational changes with the α-subunit, particularly the αIII domain and more specifically the S2B and Mo-homocitrate sites of the cofactor, as well as residues α-Gln191, α-Trp253, α-His274, α-Phe300, α-His362, and α-His451.

### 3.5. α-Subunit Flexibility and Cooperativity

As underscored above, crystallographic structural studies are limited by their requirements for highly pure, concentrated, isomorphous protein states which hinder the capture of low abundance intermediates. The crystal lattice may further restrict the landscape of allowable structural states. Single-particle cryoEM does not require similarly high protein concentrations or amounts and may tolerate sample heterogeneity; however, to achieve sufficient resolution to interpret structural changes within the active site, many thousands of identical particles must be averaged. Despite increasingly powerful classification algorithms, this processing pipeline still favors the determination of dominant states within the dataset. Taking into account these experimental limitations of structural studies, it is clear that a complete understanding of the nitrogenase reaction mechanism requires input from complementary biochemical, spectroscopic, and computational techniques. We have emphasized the following overlapping observations within mutagenized and turnover-relevant states of nitrogenase: conformational changes with the α-subunit, the S2B and Mo-homocitrate sites of the cofactor, and residues α-Gln191, α-Trp253, α-His274, α-Phe300, α-His362, and α-His451 numbered according to the *A. vinelandii* MoFe-protein sequence. In this section, we will review biochemical, spectroscopic, and computational evidence for the role of these sites in nitrogen reduction.

The possible flexibility inherent within the αIII domain of the MoFe-protein had first been structurally recognized in the crystal structure of the MoFe^∆nifB^ protein [118]. In that study, the authors suggested that the αIII domain changes conformation to reveal a positively charged funnel for insertion of the negatively charged FeMo-cofactor:*R*-homocitrate complex. Prior work by Robinson et al. has shown that MoFe-protein isolated from ∆*nifH* strains of *A. vinelandii* exhibited smearing on native PAGE gels, with the center peak migrating higher than the wild-type protein [124]. When the ∆*nifH* MoFe-protein was reconstituted with FeMo-cofactor, the gel band sharpened and migrated as wild-type. Based on these results, the authors concluded that the “*FeMo cofactor has a significant role in maintaining the structure of wild-type MoFe-protein and that when it is absent…a mixture of conformations exists in solution*”. This statement has now been bolstered not only by the MoFe^∆nifB^ structure, but also the cryoEM structures of the MoFe^Alkaline-inactivated^ and MoFe^∆nifV^. The latter structures also exhibit perturbations within the FeMo-cofactor site and corresponding disordering of the α-subunit.

Robinson et al. also speculated that the Fe-protein:ATP complex with MoFe-protein may stabilize an active site exposed state of the ∆*nifH* MoFe-protein, and that this might also be true for the wild-type protein under turnover conditions [125]. Interestingly, the cryoEM structures described by Rutledge et al. from reaction mixtures correspond to 1:1 complex states of the MoFe-protein and Fe-protein [15]. In contrast to the proposal by Robinson et al. [125], that the Fe-protein stabilizes an active site exposed state of the MoFe-protein, Rutledge et al. [15] note disordering in the α-subunit that is distal to the bound Fe-protein. In the latter study, the authors suggested that the observation of 1:1 complexes could reflect the role of half-reactivity and negative cooperativity between the two halves of the MoFe-protein during turnover, only allowing for the binding of one Fe-protein at a time. Asymmetry in the flexibility of the α-subunit has now been observed in four distinct structures, including both turnover structures of Rutledge et al. [15], as well as the MoFe^Alkaline-inactivated^ and MoFe^∆nifV^ states [111]. It remains unclear, however, what the catalytic relevance of these observations might be, including whether they could potentially be an artefact of single-particle cryoEM sample preparation reflecting asymmetric interaction of the protein with the air–water interface during blotting and freezing. Irrespective of the mechanism, disordering observed in the αIII domains of the MoFe^∆nifB^ crystal structure supports the relevance of the observed disorder in the cryoEM structures.

If not a consequence of the experimental protocol (sample handling and grid preparation), the exclusive isolation of 1:1 complex structures from the turnover mixture seems surprising, as several groups have presented evidence that the 2:1 MoFe-protein:Fe-protein complex is an active state [126,127,128,129] in addition to the 1:1 complex [16,130]. As one diagnostic for half of sites reactivity is that the binding of one inhibitor can inactivate both active sites [131,132], it is also relevant that the preparation of the ADP-AlF_4_^−^-stabilized complex of *A. vinelandii* nitrogenase proteins under turnover conditions was reported to produce 2:1 complexes [133], while a detailed study of the ADP-AlF_4_^−^ stabilized complex of the *K. pneumoniae* nitrogenase proteins indicated that the two binding sites on the MoFe-protein have the same affinity for Fe-protein, irrespective of whether one or both sites are unoccupied [134]. Counterbalancing these observations, however, are reports of long-range interactions between the two Fe-protein binding sites [16,17]. As noted above, the Lowe–Thorneley model established that substrate binding or product dissociation only occurs when the Fe-protein has dissociated from the MoFe-protein, so a role for these long-range interactions may be to promote Fe-protein dissociation under the high protein concentrations of the cell.

The only contacts between the two αβ dimers are through the two-fold-symmetry-related β-subunits. Aside from known changes in residues around the P-cluster in different oxidation states, few conformational changes have been noted within the β-subunit, obfuscating hypothesized routes of cooperativity. The particular lability of the αIII domain is of interest, as it has long been established by the first sequencing and crystallography results that the α- and β-subunits of the MoFe-protein contain homologous domains [39,135]. The equivalent βIII domain laterally spans the β-subunit extending between the dimers (Figure 8). Given the propensity for rearrangements in the αIII domain, perhaps it is possible that related rearrangements could occur within the βIII domain that may allow for communication across dimers; evidence for this has not been observed, however.

### 3.6. Probing the Importance of Structurally Malleable Residues

Ten active site amino acids appear to be strictly conserved in the α-subunit of nitrogenase: Val70, Gln191, His195, Cys275, Arg277, Ser278, Gly356, Phe381, Gly424, and His442 [136]. As described in the sections above, several structures have revealed distinct changes in residues outside of the active site, including α-Trp253, α-His274, α-Phe300, α-His362, and α-His451 (Table 1) [15,91,111,118]. Investigation of sequence conservation outside of the active site in a select subset of diazotrophs reveals that of these residues, only α-His362 is conserved, being present in greater than 90% of sequences compared [137]. Mutagenesis of this residue, as well as α-His274 and α-His451, has been observed to result in decreased cofactor accumulation on the MoFe-protein, indicating that these residues may play a role in cofactor insertion [138,139]. This is supported by the observed conformational changes in these residues within the MoFe^∆nifB^ structure, which also contains flipped rotamers of α-Trp253 and α-Phe300 relative to the resting state [118]. These results suggest that changes in these residues are correlated with movements of the cofactor or α-subunit order. Interestingly, in structural studies of *A. vinelandii* MoFe-protein crystals pressurized with xenon, a Xe binding site was observed between α-Trp72 and α-Trp253 [140]. The latter residue may then play multiple roles in both biosynthesis and in catalysis through control of substrate ingress to the active site; however, further studies are necessary to fully understand its functions. Within the active site, Sippel et al. observed a rotamer flip of α-Gln191, which they noted opened a site for sulfide binding, proposed to temporarily house the displaced S2B sulfur [48]. Mutations at this site ablate nitrogen reduction activity [141]. Interestingly, DFT studies using a 486 atom model found that while severance of the Fe2-S2B bond is feasible, complete dissociation of S2B is not [142]. In contrast to these studies, S2B is apparently completely dissociated within the CO- and Se-bound crystal structures; however, this could reflect the different dynamics between the time scales of substrate reduction and crystal growth [43,44,45,46,49,143].

Computational and mutagenesis studies have pointed to the importance of several residues that have not been observed to dramatically change in structures to date. The residue α-His195 is of particular interest due to its proximity to S2B and its ability to engage in hydrogen bonding at this site (Figure 9). Residue α-Arg96 is also poised near the cofactor and may provide a hydrogen bonding partner for the belt sulfurs if the cofactor is rearranged during turnover. Substitutions at this site have been seen to affect substrate binding [32,40]. α-Arg359 and α-Phe381 have also been proposed to line a substrate access channel to the FeMo-cofactor. Substitution at α-Arg359 with Gln significantly decreased metal content of the protein and ablated activity, while substitution at α-Phe381 was proposed to block substrate interaction with the FeMo-cofactor [144]. The large number of potential hydrogen bonding partners within the active site, as well as the multimetallic nature of the cofactor, may point to the possibility for multiple reaction mechanisms, particularly for different substrates.

## 4. Conclusions

In recent decades, discoveries of sites for ligand binding, belt sulfur lability and interchange, and hydride formation have informed our current mechanistic understanding for enzymatic nitrogen fixation by primarily focusing on events at and within the trigonal prismatic arrangement of irons in the active site cofactor. However, emerging cryoEM structural studies of turnover-relevant states have muddied the waters by revealing changes about the Mo-homocitrate site and the greater α-subunit region [15,111]. These studies re-emphasize the importance of the entire cofactor—and protein—by highlighting the role of conformational dynamics in the catalytic mechanism. The recognition that both the cofactor and the protein are dynamic under turnover suggests that more highly reduced forms may differ in key ways from the resting state structure. A further implication may well be that the high-resolution structural studies essential to unambiguously characterizing the interaction of small ligands to electron dense clusters will be compromised by the conformational dynamics associated with catalysis, in addition to the heterogeneous population of E_n_ states appearing under turnover.

The turnover and turnover-adjacent cryoEM structures have consistently exhibited asymmetric disordering within domain III of the α-subunit (Figure 4, Figure 5 and Figure 7). As discussed in the sections above, this supports the intriguing hypothesis that the nitrogenase MoFe-protein active sites may act in an alternating fashion. If true, this would suggest that reaction within one dimer induces a conformational change in the adjacent dimer [145]. A model of negative cooperativity between the active sites was proposed by Danyal et al. based on kinetic data measuring electron transfer, ATP hydrolysis, and phosphate release under pre-steady-state conditions [16]. The authors of that study further used coarse-grained modeling of the MoFe-protein:Fe-protein complex in various nucleotide-bound states. Their simulations suggested an “out-of-phase rocking” motion of the Fe-protein. These two αβ subunit pairs are only in contact through the MoFe-protein β-subunits; thus, cross-talk must occur through these subunits. As an example, in the case of the MoFe^Alkaline-inactivated^ cryoEM structure, alignment of the β-subunits with the MoFe^As-isolated^ resting-state cryo-EM structure yields an RMSD of just 0.14 Å; furthermore, no changes in either β-subunit are observed at the interfaces between the opposing α- and β-subunits. It is possible that the β-subunit only assumes a differing conformation when the Fe-protein is bound. However, no changes at these interfaces are observed in the turnover structures from Rutledge et al. [15]. Thus, it remains unclear how interactions at one active site of the MoFe-protein may control the activity of the opposing active site. The cryoEM evidence for half-of-sites reactivity assumes that the structures are mechanistically relevant; we note that interpretations of this asymmetric phenotype are further convoluted by the tendency for the αIII domain of the MoFe-protein to adhere to and unfold at the air–water interface during sample freezing for cryoEM [111]. Further studies are necessary to elucidate the potential mechanism and role of inter-subunit cross-talk within nitrogenase.

The advent of cryoEM promises exciting new discoveries about the nitrogenase mechanism, but the resolution has so far remained limited compared to routine crystallography of nitrogenase. This situation is complicated by the dynamic nature that has been revealed within the current set of nitrogenase cryoEM structures. Thus, moving forward, ground truth studies with this methodology using well-characterized states, such as the CO-inhibited structure, will be important to understand and overcome the limitations to obtain higher resolution structures for the range of species present in a sample. Advances in cryo-EM methodology, in concert with complementary computational, spectroscopic, biochemical, kinetic, and model compound studies, are driving an unprecedented ability to define the nitrogenase mechanism in both space and time. These developments fuel our optimism that we are closing in on the answer to a critical question of the nitrogenase mechanism: what happens to bound dinitrogen in the later stages of the Lowe–Thorneley cycle?

## Figures and Tables

**Figure 1 molecules-28-07952-f001:**
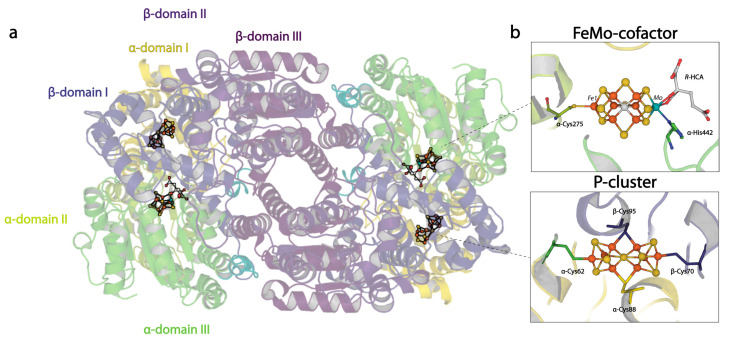
Overview of the nitrogenase MoFe-protein structure. (**a**) Quaternary structure of the nitrogenase MoFe-protein heterotetramer. α-subunits are shades of green and yellow; β-subunits are shades of blue and purple. (**b**) Metallocluster architecture of the active site FeMo-cofactor (**top**) and the P-cluster (**bottom**; PDB code 3U7Q).

**Figure 2 molecules-28-07952-f002:**
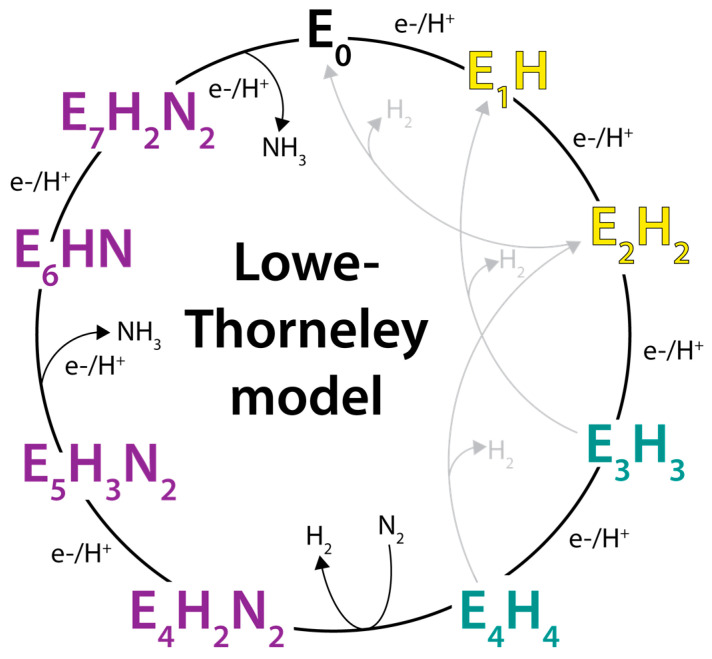
The Lowe–Thorneley kinetic model for nitrogen reduction by nitrogenase. Yellow states are competent for binding of certain alkyne substrates including C_2_H_2_. Teal states are competent for N_2_ binding. Purple states are steps at which the N_2_ triple bond may be reduced. Alternative schemes to the depicted sequence of H addition and NN bond cleavage are possible.

**Figure 3 molecules-28-07952-f003:**
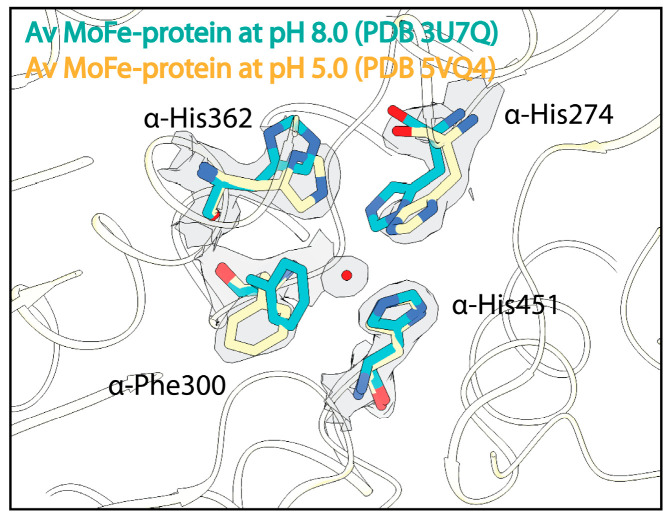
Effects of acidic pH on the structure of the nitrogenase MoFe-protein. Structural overlay of the *A. vinelandii* (Av) MoFe-protein solved at pH 8.0 in cyan and at pH 5 in yellow. Density from the Coulomb potential density map is shown in gray and waters are shown as red spheres.

**Figure 4 molecules-28-07952-f004:**
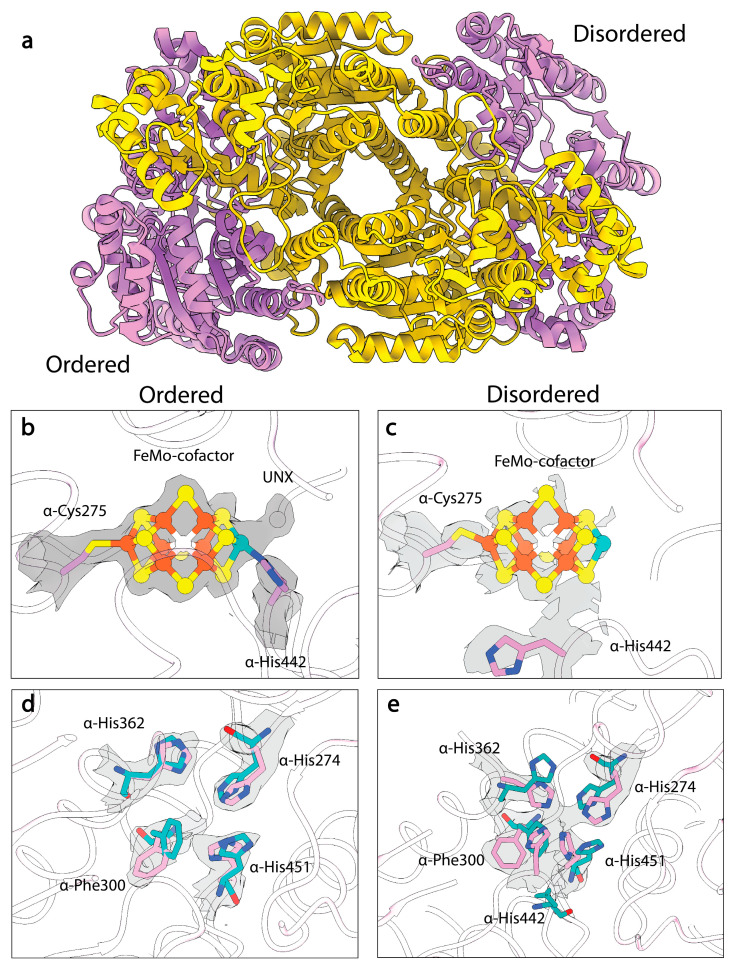
CryoEM structure of MoFe^Alkaline-inactivated^ protein. (**a**) Structural model of the MoFe^Alkaline-inactivated^ protein with α-subunits in pink and β-subunits in yellow. (**b**) CryoEM density of the FeMo-cofactor in the ordered αβ dimer of MoFe^Alkaline-inactivated^. (**c**) CryoEM density of the FeMo-cofactor in the disordered αβ dimer. (**d**) CryoEM density of the altered residues in the ordered αβ dimer of the MoFe^Alkaline-inactivated^ state (pink) compared to the resting state crystal structure (cyan). (**e**) CryoEM density of the altered residues in the disordered αβ dimer. Density from the Coulomb potential map is shown in gray. Within the FeMo-cofactor irons are shown in orange, sulfurs are shown in yellow, and the molybdenum is shown in teal.

**Figure 5 molecules-28-07952-f005:**
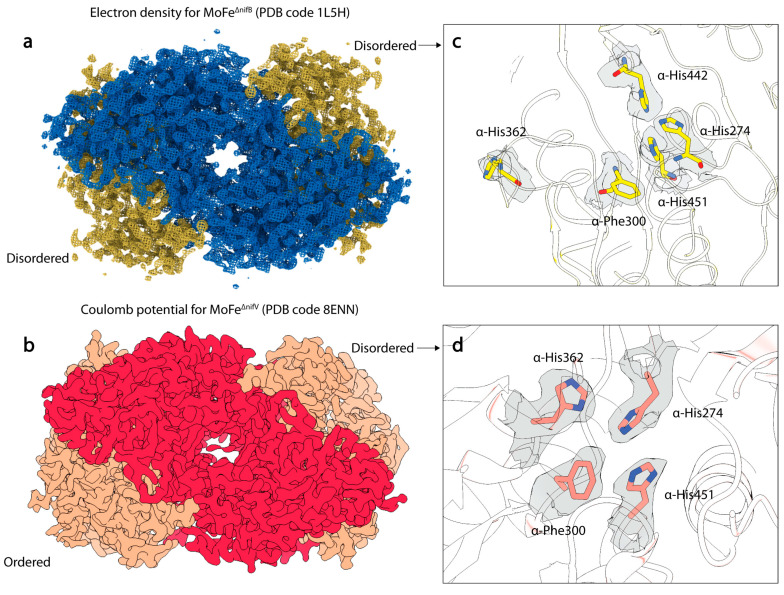
Structures of MoFe-protein from ∆*nifB* and ∆*nifV A. vinelandii* deletion strains. (**a**) Electron density of the MoFe^∆nifB^ MoFe-protein crystal structure (PDB code 1L5H). α-subunits are shown in yellow, and β-subunits are shown in blue. (**b**) Coulomb potential map (shown in gray) of the MoFe^∆nifV^ MoFe-protein single particle cryoEM structure (PDB code 8ENN). α-subunits are shown in salmon, and β-subunits are shown in red. (**c**) Electron density (show in gray) around rearranged histidine and phenylalanine residues in the α-subunit of MoFe^∆nifB^. (**d**) Coulomb potential density around rearranged histidine and phenylalanine residues in the α-subunit of MoFe^∆nifV^ in the disordered dimer.

**Figure 6 molecules-28-07952-f006:**
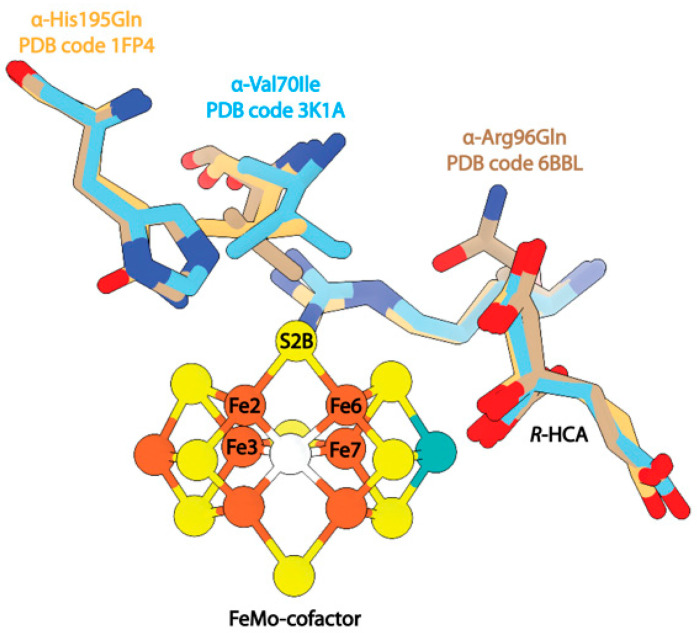
Structures of MoFe-protein single point mutants. Shown in beige is the mutant α-His195Gln (PDB code 1FP4). Shown in blue is the mutant α-Val70Ile (PDB code 3K1A). Shown in brown is the mutant α-Arg96Gln (PDB code 6BBL).

**Figure 7 molecules-28-07952-f007:**
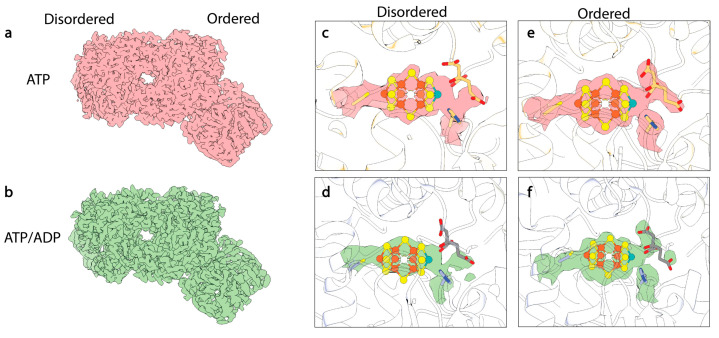
CryoEM structures of turnover states. (**a**) CryoEM map of ATP-bound 1:1 turnover complex of MoFe-protein:Fe-protein. Density from this Coulomb potential map is shown in pink. (**b**) CryoEM map of mixed ATP/ADP-bound 1:1 turnover complex of MoFe-protein:Fe-protein. Density from this Coulomb potential map is shown in green. (**c**,**d**) CryoEM density for FeMo-cofactor within the disordered αβ dimer. (**e**,**f**) CryoEM density for FeMo-cofactor within the ordered αβ dimer (see reference [15]).

**Figure 8 molecules-28-07952-f008:**
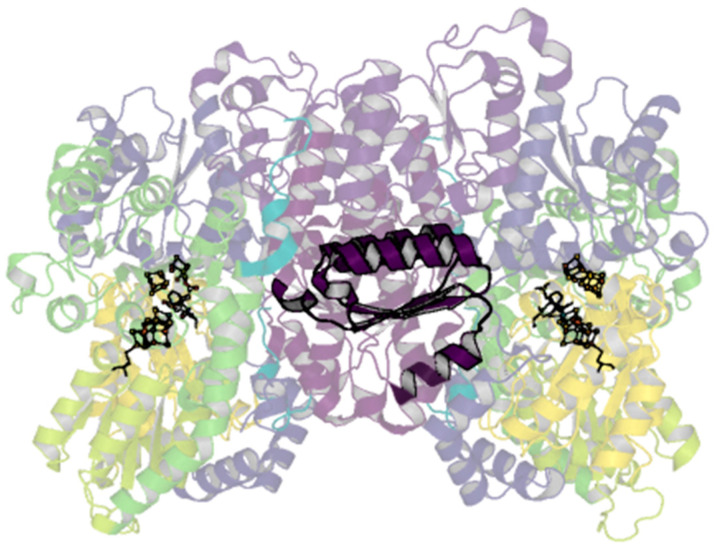
βIII domain of the *A. vinelandii* nitrogenase MoFe-protein. Structure of the nitrogenase MoFe-protein heterotetramer (PDB code 3U7Q). α-subunits are shades of yellow to green; β-subunits are shades of blue to purple. The cartoon is shown transparently with the exception of the βIII domain shown in dark purple.

**Figure 9 molecules-28-07952-f009:**
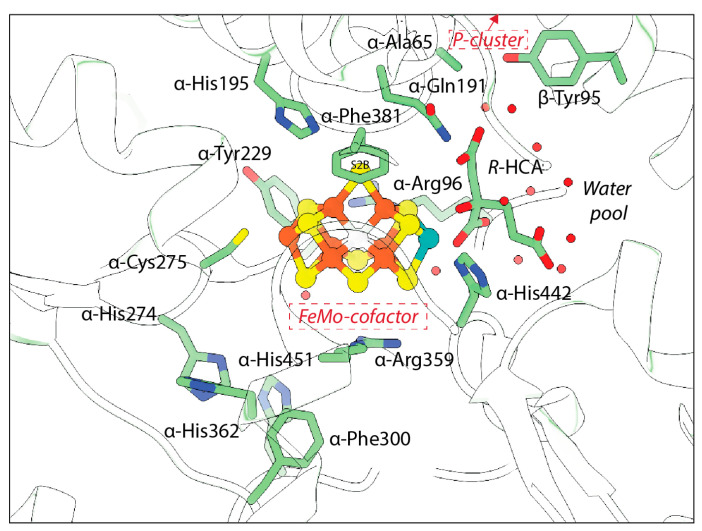
Polypeptide environment of the nitrogenase active site. The structure of the nitrogenase MoFe-protein is shown in green (PDB code 3U7Q). Side chains with known conformation changes and those near the active site are shown and labeled.

**Table 1 molecules-28-07952-t001:** Common changes observed in *A. vinelandii* MoFe-protein structures. Cells highlighted in red indicate a deviation from the resting state (MoFe^As-isolated^) structure (PDB code 3U7Q). Flip indicates an alternate conformation is observed. N/C stands for no change.

	MoFe^As-isolated^ PDB: 3U7Q	MoFe^∆nifB^ PDB: 1L5H	MoFe^∆nifV^ PDB: 8ENN		MoFe^Alkaline-inactivated^ PDB: 8ENL		^t/o^Complex 1 PDB: 7UT8		^t/o^Complex 2 PDB: 7UT9		MoFe^Acidic^ PDB: 5VQ4	MoFe^CO^ PDB: 4TKV	MoFe^CO-CO^ PDB: 7JRF	MoFe^Se^ PDB: 5BVG
Cα RMSD with 3U7Q		0.370	0.385		0.387		0.405		0.433		0.257	0.101	0.090	0.077
Asymmetric	No	No	Yes		Yes		Yes		Yes		No	No	No	No
Disordered	No	Yes	Yes		Yes		Yes		Yes		No	No	No	No
Ordered/Disordered dimer			Ord.	Dis.	Ord.	Dis.	Ord.	Dis.	Ord.	Dis.				
α-Gln191	Reference	Flip	N/C	Flip	N/C	Flip	N/C	N/C	N/C	N/C	N/C	N/C	N/C	N/C
α-Trp253	Reference	Flip	N/C	Flip	N/C	Flip	N/C	Flip	N/C	Flip	Flip	N/C	N/C	N/C
α-His274	Reference	Flip	N/C	Flip	Flip	Flip	N/C	Flip	N/C	Flip	Flip	N/C	N/C	N/C
α-Phe300	Reference	Flip	N/C	Flip	Flip	Flip	N/C	Flip	N/C	Flip	Flip	N/C	N/C	N/C
α-His362	Reference	Flip	N/C	N/C	N/C	Flip	N/C	N/C	N/C	N/C	Flip	N/C	N/C	N/C
α-His451	Reference	Flip	N/C	Flip	Flip	Flip	N/C	Flip	N/C	Flip	N/C	N/C	N/C	N/C
β-Gln93	Reference	Flip	N/C	N/C	N/C	Flip	N/C	N/C	N/C	N/C	N/C	N/C	N/C	N/C

## Data Availability

Data are available in original cited literature and corresponding supporting information. Additionally, all structures discussed in this review are publicly available through the Research Collaboratory for Structural Bioinformatics Protein Data Bank.

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
