# Peer review of "Nitrogenase beyond the Resting State: A Structural Perspective"

_molecules, 2023, doi:10.3390/molecules28247952_

Round 1
Reviewer 1 Report
Comments and Suggestions for Authors
This review from Warmack and Rees brings readers up to speed on understanding how to understand the most recent results in the structural biology of the Mo nitrogenase in the context of the decades of prior research on this fascinating enzyme. The manuscript offers a masterclass in review writing: it is provocative but measured, it offers opinions while clearly stating them as such and not dismissing alternative views, and it emphasizes current work and future directions. It will be great reading for newcomers to this area and to seasoned investigators, to structural biologists and microbiologists as well as to inorganic and theoretical chemists.
A few suggestions:
1. Since computational studies are mentioned in the context of identifying possible sites/states of protonation at FeMo-co, it may be appropriate to mention work from groups in addition to Ryde’s (e.g., Bjornsson, Raugei, Seigbahn). Probably only work that entails sulfide displacement is worth considering nowadays.
2. Regarding the loss of homocitrate at high pH, it is concluded that “The disappearance of this functional group under high pH turnover conditions suggests that a deprotonation event occurs which cannot be reversed due to the limited concentration of protons, resulting in an irreversible rearrangement leading to the loss of homocitrate from the protein.” This suggestion is perfectly reasonably, but it is also possible that the loss of homocitrate is due not solely to the absence of protons but also to the presence of hydroxide, which can act as a nucleophile and displace homocitrate at Mo.
Author Response
We are grateful for this reviewer's supportive and constructive feedback. We have modified the manuscript in the following ways:
- In response to Reviewer 1's comment: Since computational studies are mentioned in the context of identifying possible sites/states of protonation at FeMo-co, it may be appropriate to mention work from groups in addition to Ryde’s (e.g., Bjornsson, Raugei, Seigbahn). Probably only work that entails sulfide displacement is worth considering nowadays.
- We have significantly expanded our discussion of computational studies by adding section 2.4.3 "Computational studies of the FeMo-cofactor beyond the resting state" which highlights the work be Bjornsson, Raugie, and Seigbahn.
- In response to Reviewer 1's comment: Regarding the loss of homocitrate at high pH, it is concluded that “The disappearance of this functional group under high pH turnover conditions suggests that a deprotonation event occurs which cannot be reversed due to the limited concentration of protons, resulting in an irreversible rearrangement leading to the loss of homocitrate from the protein.” This suggestion is perfectly reasonably, but it is also possible that the loss of homocitrate is due not solely to the absence of protons but also to the presence of hydroxide, which can act as a nucleophile and displace homocitrate at Mo.
- We thank the reviewer for pointing out the limits of our interpretation. We have added a statement to the section discussing this structure as follows: "Alternatively, the elevated concentration of hydroxide ions in the alkaline solution may result in nucleophilic attack that displaces the homocitrate."
Reviewer 2 Report
Comments and Suggestions for Authors
This is a quite impressive review of the current status of the nitrogenase field. The authors are especially to be commended for the very even-handed way in which they cover several quite controversial that have appeared in the last few years. Coverage of the literature is comprehensive and effectively organized along thematic lines in such a way that makes the review extremely easy to read. As far as this reviewer is concerned, the manuscript is ready to publish aspis.
Author Response
We greatly appreciate Reviewer 2's supportive comments.
Reviewer 3 Report
Comments and Suggestions for Authors
Warmack and Rees have submitted a review which summarizes established knowledge and then focusses on fairly recent insights into the operation of nitrogenase(s) gained in particular via X-ray structural and cryoEM studies that address mutated enzymes as well as reactive intermediate state structures. The authors start with a more general brief introduction and context of nitrogen fixation before getting to their actual topic. First the well investigated resting state and earlier work on few of the much more elusive transition states are discussed including structures, binding and reactivity models as the current state-of-the-art. Afterwards some fascinating recent insights are portrayed which further our knowledge of what is going on in these critically important enzymes without having really resolved everything as of yet. However, this dense review will get the colleagues thinking who work on nitrogenase on a broader as well as deeper level. The results of more recent and challenging studies are eloquently presented and also assessed comprehensively in their context. Difficulties and potential stumbling blocks are pointed out.
It was really a very good read and reviewing this piece of writing was quite pleasurable. The text will be exceedingly useful for anyone teaching bioinorganic chemistry and it will most certainly receive a very high number in citations from the community. The review will be a valuable asset of the journal Molecules and it is recommended accepting the manuscript after a very minor revision.
Issues to be considered are almost exclusively typos. It appears most convenient pointing these out directly within the pdf-copy for review. Said file is attached and includes also one or the other comment with suggestions that might improve the respective sentence. N.B.: to see the comments, the file likely must be opened with Adobe or similar programme (not in the browser).

Author Response
We thank Reviewer 3 very much for their thorough analysis of our manuscript, especially in regards to identifying typos and mistakes, as well as their supportive comments.
In response, we have gone through and fixed all typos identified by Reviewer 3 and altered the reference format so that all authors are listed.